# Evaluation of opportunities to implement community-wide mass drug administration for interrupting transmission of soil-transmitted helminths infections in India

Kumudha Aruldas[1‡], Kim Dawson[2‡], Malvika Saxena[1], Angelin Titus[1], Jabaselvi Johnson[1], Marie-Claire Gwayi-Chore[2], Jayaprakash Muliyil[1], Gagandeep Kang[1], Judd L. Walson[3,4], Ajay Khera[5], Sitara S. R. Ajjampur🄳[1]*, Arianna Rubin Means[2,4]

1 The Wellcome Trust Research Laboratory, Division of Gastrointestinal Sciences, Christian Medical College Vellore, Vellore, Tamil Nadu, India, 2 Department of Global Health, University of Washington, Seattle, Washington, United States of America, 3 Departments of Global Health, Medicine (Infectious Disease), Pediatrics and Epidemiology, University of Washington, Seattle, Washington, United States of America, 4 The DeWorm3 Project, University of Washington, Seattle, Washington, United States of America, 5 Ministry of Health and Family Welfare (former), Government of India, New Delhi, India

‡ Co-lead authors
* sitararao@cmcvellore.ac.in

## Abstract

### Background

The World Health Organization Neglected Tropical Disease (NTD) guidelines recommend control of soil transmitted helminth (STH)-associated morbidity with targeted deworming of preschool and school-aged children who are disproportionately affected by STH-associated morbidity. However, this strategy leaves many adults untreated and reinfection within communities perpetuates transmission even when mass drug administration (MDA) coverage of children is high. Evidence suggests that it may be possible to interrupt STH transmission by expanding MDA to a community-wide MDA (cMDA).

### Methods

This multi-methods study of organizational readiness survey, key informant interviews, and program mapping, were conducted with government stakeholders in three Indian states, Goa, Sikkim, and Odisha, to assess readiness of the states for transitioning from school-based MDA to cMDA and identify opportunities to leverage existing infrastructure from other NTD programs like lymphatic filariasis (LF) for STH cMDA.

### Principal findings

Overall, all three states indicated a highly favorable policy environment, effective leadership structure, adequate material resources, demonstrated technical capacity, and adequate community infrastructure needed to launch a STH cMDA program. The findings indicated a high-level of health system readiness to implement provided human resources and financial

**Data Availability Statement:** All relevant data are within the manuscript and its supporting information files.

**Funding:** The study received funding from Child Investment Fund Foundation (R-1707-01983, PI JLW). The funders had no role in the study design, data collection and analysis, decision to publish or preparation of the manuscript.

**Competing interests:** The authors have declared that no competing interests exist.

resources to deliver cMDA is strengthened. Areas with a significant overlap between LF and STH MDA platforms, particularly at the community-level, may be best primed for transitioning. Immunization, maternal child health, and non-communicable disease control programs were the other programs for possible integration of cMDA. States indicated having effective leadership structures in place at the state-level, however, engaging local leaders and community groups were considered crucial for successful implementation of cMDA. In-migration was a perceived challenge for estimating drug requirement and preventing possible stockouts.

## Conclusions

Findings from this study are intended to proactively support government decision making, prioritization, and program planning across heterogenous implementation contexts in India to speed the translation of research findings into practice.

## Clinical trial registration

NCT03014167; ClinicalTrials.gov.

### Author summary

Soil-transmitted helminths (STH) are highly prevalent neglected tropical diseases globally. Current WHO recommendations for the control of STH focus on treating children, women of childbearing age, and others with occupations at high risk for STH infections such as agriculture. Evidence from mathematical models suggests that it may be possible to interrupt STH transmission by treating everyone in the community through community-wide mass drug administration (cMDA). This multi-methods study was conducted with government stakeholders in three Indian states, Goa, Sikkim, and Odisha, to assess their readiness for transitioning to a cMDA program for STH. Three data collection methods, organizational readiness surveys, key informant interviews, and program mapping were used in this study. The results showed a high-level of health system readiness in all the three states with respect to policy environment, leadership structure, material resources, technical capacity, and community infrastructure to implement a cMDA program. Stakeholders reported the potential for possible integration of cMDA with other programs such as for lymphatic filariasis, maternal and child health, and non-communicable diseases provided the human and financial resources are strengthened, and local leaders and community groups are engaged in the program. In-migration was a perceived challenge for estimating drug requirement and preventing possible stockouts.

## Introduction

Soil transmitted helminths (STH) infect an estimated 1.45 billion people globally [1]. India has the highest burden of STH, with approximately 25% of the global burden [2–5]. STH are associated with inadequate access to water, sanitation, and safe living conditions [6]. At moderate to high infection intensities, STH are associated with a significant morbidity, particularly in children, including anemia, malnutrition, chronic malaise, impaired cognitive development, poor school performance and stunting [7]. In addition, STH infections in pregnant women are

associated with intrauterine growth retardation and low birth weight babies [8–10]. The World Health Organization (WHO) STH guidelines recommend controlling STH-associated morbidity with mass drug administration (MDA) of albendazole or mebendazole for pre- and school age children, women of childbearing age, or people with high-risk occupations (e.g., agriculture) [11]. In India, teachers, Accredited Social Health Activists (ASHAs), and Anganwadi Workers (AWWs) administer deworming treatments in schools during National Deworming Days (NDDs), treating an estimated ~240 million children biannually [12].

There are several challenges influencing the effectiveness and sustainability of school-based deworming. For example, some children do not attend school and thus may not have access to deworming [13]. Additionally, because adults are not broadly treated for STH, they can serve as reservoirs of infection in the community [13,14]. Lastly, as school-based MDA would not interrupt transmission especially of hookworm, these programs might need to endure indefinitely to continue suppressing STH-related morbidities amongst pediatric and high-risk populations [15]. As a result, current programs, which are heavily reliant on global drug donations, would likely need to continue for the long term [15,16].

Mathematical models suggest that transitioning from school-based MDA to community-wide MDA (cMDA) could potentially address inequities in MDA treatment coverage and reduce the number of years needed to deliver school-based treatment at scale by treating adult reservoirs of disease in the community, leading to STH transmission interruption [15,17]. The strategy of cMDA with high treatment coverage has been successful in interrupting transmission for NTD programs, such as lymphatic filariasis (LF) [18]. Of the 72 countries endemic for LF in 2017, 17 eliminated LF as a public health problem and are monitoring for recrudescence [18]. In India, approximately 40% of LF endemic districts have successfully entered the post-MDA surveillance phase [18]. The cMDA infrastructure delivery platform created for LF programs in India provide opportunity to implement STH cMDA in areas which are co-endemic for LF and STH. Both STH and LF programs include MDA of albendazole, as a result, concomitant reductions in STH prevalence are typically observed in LF programs [16]. However, as LF programs transition to a post-MDA surveillance phase, rebounds in STH infections may be expected. There is an opportunity for strategic de-implementation of LF programs, meaning LF activities and infrastructure can be leveraged and repurposed for delivery of STH cMDA. New STH cMDA programs may benefit from access to established LF platform resources including trained health workers, supply chains, and community engagement mechanisms. An STH cMDA platform may also provide the infrastructure necessary for ongoing LF surveillance and completion of WHO elimination dossiers [16].

The feasibility of STH transmission interruption by cMDA with a community-based cluster randomized trial, the DeWorm3 study, is ongoing in India (in the state of Tamil Nadu), Benin and Malawi and is in the surveillance phase [14]. In 2018, we carried out a qualitative research with government stakeholders to assess readiness for transitioning to cMDA for STH in India, and to identify strategic opportunities to leverage existing LF infrastructure. This multi-methods study with government stakeholders was carried out in three Indian states, Goa, Sikkim, and Odisha. Findings from this study are intended to proactively support government decision-making regarding scale down of LF platforms, prioritization of cMDA for STH, and program planning across heterogenous implementation contexts.

## Methods

### Ethics statement

The Institutional Review Board of Christian Medical College Vellore (10392 [INTERVEN]), and the Human Subjects Division at the University of Washington (STUDY00000180)

approved the DeWorm3 trial in India. The trial is registered at ClinicalTrials.gov (NCT03014167). The CMC IRB approved this study as an amendment (IRB–A4, September 26, 2018). Written informed consent was obtained from all who participated in the study.

## Study location and setting

This multi-methods study was conducted in three Indian states: Goa, Sikkim, and Odisha. The states were selected using a systematic process to identify states with high potential for STH transmission interruption through cMDA. NTD stakeholders at State Ministry of Health & Family Welfare (State MOHFW) offices participated in three primary methods of data collection, including organizational readiness surveys, key informant interviews, and program mapping exercises, to assess readiness to transition from school-based MDA to cMDA for STH in India.

## Study design and data collection

A state selection tool was developed and included a multi-criteria decision analysis (MCDA) with six categories: STH epidemiologic profiles, helminth associated morbidity, presence of helminth transmission risk factors, community wide public health coverage indicators, and human resource capacity to engage in intensified STH programming. The purpose of this tool was to select three states from diverse Indian regions (north, south, east/northeast, and west/central), with a high potential for STH transmission interruption. All states and territories were scored and scores were shared during a national-level meeting at National Institution for Transforming India (NITI) Aayog in October 2018 with government stakeholders (Child Health division, Women and Child Health department, National Centre for Disease Control, Indian Council of Medical Research, and members of NITI Aayog) and non-governmental technical support partners (Nutrition International, the Bill & Melinda Gates Foundation, UNICEF, Child Investment Fund Foundation, All India Institute of Medical Sciences, and Christian Medical College, Vellore). During this meeting, three states were selected from three regions: Goa (West/Central), Odisha (East), and Sikkim (Northeast). Goa and Odisha are LF endemic states, with a mix of districts either actively conducting MDA or already in a post-MDA surveillance phase. The study population included the government officials responsible for state-level NTD decision making and implementation in Goa, Sikkim, and Odisha. They were purposefully selected from officers and decision makers directly responsible to training, executing, and monitoring the program. They were invited to participate in three data collection streams: organizational readiness surveys, key informant interviews, and program mapping.

## Organizational readiness survey

Organizational readiness is the degree to which members believe they are prepared to implement organizational change [19]. When readiness is high, members are more likely to initiate a change and provide persistent support to the proposed change. Lower readiness indicates the change is undesirable and there is reluctance or even resistance to implementing change [20]. The organizational readiness survey to assess states' readiness to implement cMDA consisted of 44 questions scored on a 5-point Likert scale including a response option of "not enough information to answer" (Annex 1 in S1 File). More information about survey scale development can be found elsewhere [21]. The survey items are organized by an adapted version of health system building blocks, including policy environment, leadership structure, human resources, technical capacity, material resources, financial resources, and community delivery infrastructure [22]. Medians and interquartile ranges (IQRs) were calculated for each survey

item and domain, by state. Responses of "not enough information to answer" did not contribute to median estimates.

## Qualitative interviews

All individuals who completed organizational readiness surveys were invited to participate in audio recorded key informant interviews. A semi-structured interview guide consisting of twelve questions and five probes (Annex 2 in S1 File) was developed, informed by the Consolidated Framework for Implementation Research (CFIR). The CFIR is a multi-level framework that can help systematically assess potential barriers and facilitators prior to implementing an innovation, such as transitioning to cMDA [23]. Interviews focused on questions about stakeholder perceptions of readiness to transition to cMDA as well as how cMDA for STH might leverage existing LF platforms. The 45-minute interviews were conducted in English, transcribed, and then coded using a mix of a *priori* thematic coding from a CFIR informed codebook and inductive open coding. Transcripts were independently coded by two primary coders using ATLAS.ti software 8.0 (Scientific Software Development GmbH, Berlin). The coded transcripts were finalized in consensus, with a third coder consulted when necessary.

## Program mapping

Three stakeholders in each state (n = 9) were invited to participate in a state specific program mapping exercise of NDD and, separately, LF MDA programs to solicit information on: (1) key activities involved in MDA delivery and their timing, (2) material resources and/or infrastructure needed to support activities, (3) level of government involvement within each activity, and (4) unique state-level characteristics that facilitate or challenge the implementation of that activity (Annex 3 in S1 File). The program mapping data were used to identify and understand changes in NDD program activities, personnel, and resources that may be needed if the state were to transition to STH cMDA.

# Results

From each of the three states, 15–17 government stakeholders participated in organizational readiness surveys, key informant interviews, or program mapping activities (Table 1). Results from the three data collection streams are organized and presented by the health system domain.

## Policy environment

Program mapping data indicated that India's National Deworming Day (NDD) program is primarily housed within the Department of Maternal and Child Health (MCH) while the LF program, implemented only in Goa and Odisha, resides in the Department of Vector Borne Diseases (VBD), both within the national Ministry of Health and Family Welfare (MOHFW). The timing of NDD and LF key activities and the process is described in Table 2. All three

**Table 1. Number of study participants from each state.**

| State | Organizational Readiness Survey | Key Informant Interview | Program Mapping |
|---|---|---|---|
| Sikkim | 5 | 5 | 3 |
| Goa | 6 | 6 | 6 |
| Odisha | 6 | 5 | 6 |
| **Total** | **17** | **16** | **15** |

**Table 2. Program mapping findings–Goa and Odisha.**

| Goa | | | |
| --- | --- | --- | --- |
| Activity | NDD timing leading up to implementation | LF timing leading up to implementation | Key similarities/differences |
| Drug requirement determined from population projection (adding 10% buffer) | 2 months | 3 months | • The Goa LF-MDA program is in Transmission Assessment Survey (TAS) phase.<br>• For LF, Anganwadi Workers and Multi-Purpose Health Workers prepare DEC & Albendazole requirements based on population. This estimate is submitted to Medical Officer In-Charge (MOIC)/Health Officer who send updated census data and drug requirement to the State Program Manager.<br>• For NDD, estimates are made for school and pre-school populations, not the entire state population. Similarly, MOICs-Block PHCs send their school and pre-school drug requirements to State Program Manager and State Consultant-SFWB. |
| Activity-based annual budget is prepared for Annual Program Implementation Plan (APIP) | >6 months | 6 months | • For LF the budget is submitted by the Deputy Director-NVBDCP.<br>• For NDD the budget is submitted by the Extension Educator.<br>• Both programs are reviewed and approved by MOHFW. |
| Drug procurement process initiated | 2 months | 3 months | • For LF, the State Program Manager compiles DEC and Albendazole requirement from block-level data and sends to Chief Medical Officer- Directorate of Health Services (CMO-DHS). From there CMO-DHS verifies and forwards to Deputy Director-NVBDCP to send the compiled drug requirement to MOHFW.<br>• For NDD, the State Program Manager places the drug order with Medical Store Depot to procure albendazole after approval from CMO- State Family Welfare Bureau (SFWB).<br>• The LF program requires several layers of review for placing the drug order and is ordered at a hire position level. |
| State coordination meeting held | 2 months | 5 months | |
| Implementation letter issued from MOHFW to begin program implementation | 3 months | 4 months | |
| State Entomologist conducts Microfilaria Survey before every LF-MDA round in sentinel/random sites | Not applicable | 1 month | Random site sampling/sentinel site sampling does not happen in NDD. |
| Drug quality checked | 1 month | 1 month | |
| IEC materials developed and distributed | 2 months | 3 months | • For NDD, the State Extension Educator revises the Information, Education & Communication (IEC) materials and obtains approval from CMO-SFWB/Deputy Director-DHS. The State Extension Educator also ensures translation of IEC and training materials into Konkani and Marathi (local languages). Once approved, the State Extension Educator sends IEC/training/reporting materials to the Department of Printing or issues a No Objection Certificate for printing in the open market. Printed IEC and training materials are received by the State Extension Educator approximately two weeks before drug distribution. Then, the State Consultant-SFWB alerts MOIC-Block Primary Health Center (PHC) to collect IEC materials from DHS office.<br>• Similarly, in LF, the State Extension Educator also makes state-specific modifications to the IEC material.<br>• Both programs are sensitive in making state-level modifications and translating to common languages within given state. |
| Drugs are received at State level | 1 month | 3 months | • For LF, the Deputy Director- NVBDCP receives drugs at DHS.<br>• For NDD, drugs are also received at the DHS office, but by the State Program Manager. |
| Planning for adverse events | 2 months | 2 months | |
| National level training/orientation | 1 month | 4 months | |

*(Continued)*

**Table 2.** (Continued)

| | | | |
|---|---|---|---|
| State level training/orientation | 1 month | 2 months | |
| Drugs received at block level | 2 weeks | 2 weeks | • For NDD, Block PHC Nurses collect the drugs, IEC materials and training material from DHS Office. They are supervised by the State Program Manager, Consultant-SFWB, State Nurse-SFWB. Then about 1 week before drug distribution, the Rashtriya Bal Swasthya Karyakram (RBSK) Mobile Health Team and AYUSH Doctors distribute drugs to schools/Anganwadi centers.<br>• For LF, the Deputy Director—NVBDCP issues a letter to the Senior Malaria Inspector to disburse the required number of drugs, as well as IEC materials, and reporting forms to block CHCs. The PHC Extension Educator collects these items and sends to health sub-centers. |
| Block level training/ orientation | 2 weeks | 1 month | • For LF, Block MOICs conduct training for drug distributors/community health workers.<br>• This is similar to the NDD program, except for in NDD only teachers and AWWs are trained. During training, IEC, and reporting forms are also distributed. The CMO-SFWB notifies MOIC-Block PHC to train village-functionaries, where materials are also distributed. MOIC-Block PHCs compile training reports and send them to CMO-SFWB. |
| Pre-MDA monitoring | 2 months | 2 months | |
| More localized sensitization | 1 week | 1–2 weeks | • Both programs give special attention to sensitizing at community levels. For LF, the State Extension Educator conducts mass media sensitization activities. The CMO-DHS directs Health Center MOICs to begin IEC activities, and Auxiliary Nurse Midwife (ANM)/Multi-Purpose Health Workers conduct community sensitization activities at the village/ward level.<br>• For NDD, teachers sensitize children about Albendazole in schools, while Community Health Workers distribute pamphlets to create awareness about importance of deworming during VHNDs and put-up banners in schools and health centers. Deworming messages are displayed across health facilities, district hospitals, government offices, and Panchayati Raj institutions. The State Extension Educator also sends targeted messages issued by MOHFW. The SMS reminders to functionaries at all levels help reinforce important program information. |
| Drugs distribution | Day of MDA | Day of MDA | • For LF, Multi-Purpose Health Workers conduct supervised drug administration house-to-house.<br>• For NDD, School teachers consume albendazole tablets in front of children before administering albendazole to them in school. |
| Monitoring visits | Day of MDA | Day of MDA | |
| Mop-up distribution | 1 week after MDA | 1 week after MDA | |
| Drug consumption coverage data compiled and reported up | 1–2 weeks after mop-up | 1–2 weeks after mop-up | • Both programs utilize hierarchical reporting in which data are collected from more local and reported and compiled as you move up.<br>• For LF, the Block MOIC prepares a coverage report to send to the State. From there, the Deputy Director-NVBDCP approves and sends State LF-MDA coverage report to MOHFW. Additionally, the Deputy Director-NVBDCP issues a letter to the Goa Medical College to conduct an evaluation of MDA coverage and drug compliance for the state.<br>• For NDD, the RBSK Mobile Health Team and Ayurveda, Yoga and Naturopathy, Unani, Siddha and Homeopathy (AYUSH) doctors draft the NDD report to send to MOIC-Block PHC. Next, the MOIC-Block PHC compiles coverage data and sends to Lady Medical Officer- DHS. The State Program Manager consolidates NDD coverage for State, then the Data Entry Operator- DHS enters data in the online NDD portal maintained by MOHFW. |

(*Continued*)

**Table 2.** (*Continued*)

| Activity | NDD timing leading up to implementation | LF timing leading up to implementation | Key similarities/differences |
|---|---|---|---|
| Unused tablets are returned | 1–2 weeks after mop-up | 1–2 weeks after mop-up | Both programs return unused tablets to PHCs to be stored for the next round of drug distribution. |
| **Odisha** | | | |
| **Activity** | **NDD timing leading up to implementation** | **LF timing leading up to implementation** | **Key similarities/differences** |
| Drug requirement determined from population projection (adding 10% buffer) | > 6 months | > 6 months | • For LF, this is done only for LF endemic districts.<br>• The projection is determined by Consultant-Entomologist, whose position can be appointed from various international agencies such as BMGF/USAID/WHO.<br>• For NDD, the population estimated drug requirement is sourced by UNICEF in all districts. |
| Activity-based annual budget is prepared for APIP | > 6 months | > 6 months | • For LF, the budget is prepared by a state consultant. The budget is reviewed by Entomologist and Finance consultants, as well as Mission Director before final approval by MOHFW.<br>• For NDD, budget preparation and review follow a similar process, but occurs in a different department than LF. Different departments (MCH and VBDCP) but similar paths and levels of review. |
| Drug procurement process initiated | 4 months | 4 months | • In LF, drugs are sent from central government in addition to being procured at a state level. DEC is sent from central government, whereas Albendazole is procured at the state level.<br>• For NDD, Albendazole is ordered by State Nodal officers and procured by Odisha State Medical Corporation Ltd. |
| Director-Family Welfare organizes Coordination Committee meeting under the chairmanship of State Secretary-Health | 2 months | 6 months | Coordination meeting for LF is conducted much earlier than NDD. |
| Implementation letter issued by MOHFW to begin program implementation | 2 months | 4 months | |
| AD-NVBDCP confirms LF districts to be covered during upcoming NDD round | 2 months | Not applicable | |
| Drug quality checked | 3 months | 1 month | • Drugs sent from the government are not quality controlled. However, if drugs are procured by state they are sent to Quality Check (QC).<br>• For LF, Albendazole is sent to QC (because it is state procured), however DEC (sent from government) is not.<br>• For NDD, Odisha State Medical Corporation Ltd conducts quality check of Albendazole & updates QC report on online portal. |
| IEC materials developed and distributed | 1 month | 4 months | |
| Drugs received at State level | 1 month | 2 months | |
| Second state technical meeting | 1 month | 2 months | |
| Media sensitization | 1 month | 1 months | |
| Planning for adverse events management | 1 month | 1 months | • For both LF and NDD planning for adverse events management includes dissemination of operational & financial guidelines, assigning of duties; dissemination of guidelines on adverse event management, establishing a control room.<br>• NDD involves government medical college and private medical college for pharmacovigilance support. |
| National level training/orientation | 1 month | 1 month | For both LF and NDD programs, the MOHFW organizes a national orientation training workshop for Nodal Officers on NDD implementation framework, & state-level program management guidance |
| State level training/orientation | 1 month | 1 months | |
| District level training/orientation | 1 month | 1 months | |
| Drugs & IEC materials sent to the districts | 2 weeks | 3 weeks | |
| Drugs received at block levels | 2 weeks | 2 weeks | |
| Block level training/ orientation | 1 week | 2 weeks | |

(*Continued*)

**Table 2.** (Continued)

| | | | |
|---|---|---|---|
| Pre-MDA monitoring | Not applicable | 2 weeks | For LF, the State M&E team undertakes pre-MDA monitoring visit at district and sub-district level. District MO-VBDTS also conduct monitoring visits to check availability of drugs at the health centers |
| More localized sensitization | Days leading up and day of MDA | Days leading up and day of MDA | |
| Drugs distribution | Day of MDA | Day of MDA | |
| Monitoring visits conducted | Day of MDA | Day of MDA | Both programs utilize the same workforce (ANMs, ASHAs and AWWs), however NDD also utilized school nodal teachers |
| Mop-up distribution conducted | 1 week after MDA | 1 week after MDA | For LF, mop-ups are conducted door-to-door, for NDD mop-up is conducted in the schools |
| Drug consumption coverage data compiled and reported up | 1–2 weeks after mop-up | 1–2 weeks after mop-up | Both programs utilize hierarchical reporting in which data are collected are compiled at the local level and moved up for collation at the district and state level |
| Unused tablets returned | 1 week after mop-up | 1 week after mop-up | • For LF, ASHAs & Drug Administrators (Das) return unused tablets to the PHC. From there, PHCs return unused tablets to CHC, and CHCs return unused tablets to senior pharmacist at district. The state uses left over albendazole for the next round.<br>• For NDD, districts keep tablets for next round of LF-MDA if they did not clear TAS. If cleared, they send the tablets to other district as instructed by State Entomologist. |

**Acronyms:** ANM–Auxiliary Nurse Midwife, APIP—Annual Program Implementation Plan, ASHA—Accredited Social Health Activist, AW–Anganwadi, AWW—Anganwadi Worker, AYUSH–Ayurveda, Yoga and Naturopathy, Unani, Siddha and Homeopathy, BMGF–The Bill & Melinda Gates Foundation, CHC–Community Health Center, CHW–Community Health Worker, CMO–Chief Medical Officer, DA—Drug Administrators, DEC–Diethylcarbamazine, DHS—Directorate of Health Services, IEC—Information, Education & Communication, LF—Lymphatic filariasis, MCH—Maternal and Child Health, MDA—Mass drug administration, MOHFW—Ministry of Health & Family Welfare, MSD—Medical Store Depot, NDD—National Deworming Day. MO—Medical Office, MOIC—Medical Office In-charge, NVBDCP—National Vector Borne Disease Control Programme, PHC–Primary Health Center, SFWB—State Family Welfare Bureau, STH—Soil-transmitted helminths, TAS–Transmission Assessment Survey, UNICEF—United Nations Children's Fund, USAID—The United States Agency for International Development, VBDTS—Vector Borne Disease Technical Supervisor, WHO–World Health Organization, QC–Quality Check

states exhibited high readiness for change within the policy environment (Fig 1). Stakeholders were supportive of STH cMDA and believed that it is necessary to control STH transmission and cMDA would help in controlling as adults are also dewormed. They indicated that cMDA is aligned with India's national NTD policy (median readiness score: 4.5–5, IQR: 1.1) (Table 3) and that India's National NTD Implementation Plan already provides sufficient guidance for implementing cMDA programs, such as LF. Stakeholders believed their State MOHFW can deliver cMDA with high coverage.

During key informant interviews, stakeholders were particularly enthusiastic about a potential STH cMDA policy due to the co-benefit for other priority health areas, such as reducing morbidity outcomes associated with anemia.

> *"It (deworming) is important. . . Anemia is leading to many issues in mother and child. . . . Anemia in an antenatal mother is a serious problem. . . she might die. . .. Even elderly have anemia related problems."* (Participant #3, Goa)

In key informant interviews, there were mixed opinions regarding whether cMDA could effectively interrupt STH transmission due to high rates of migration and environmental reservoirs. However, evidence from LF program maps indicated opportunities to achieve high STH cMDA coverage by, for example, tailoring programs for adult migrants by delivering treatment at construction sites (i.e., brick kilns, mining sites). Regarding environmental reservoirs, key

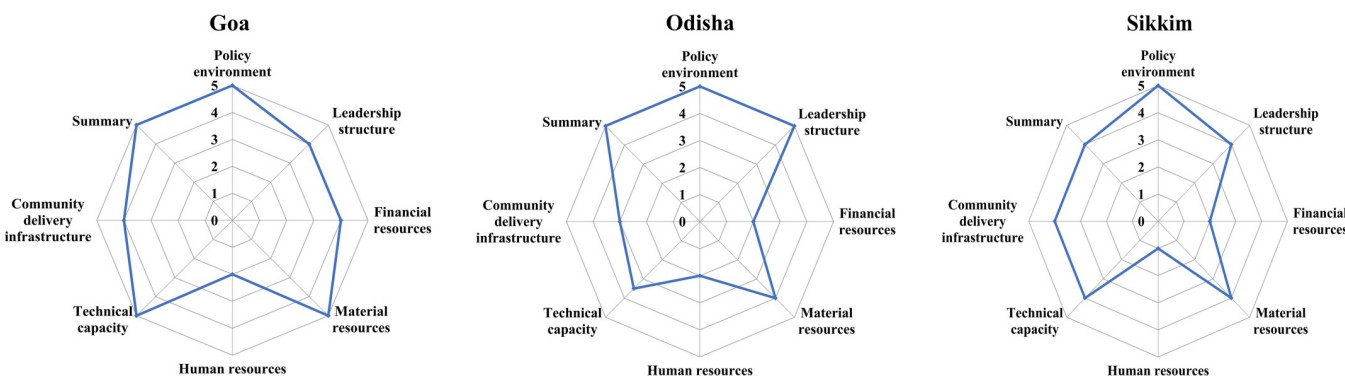

**Fig 1. Median Readiness by health system domain.**

informants strongly felt that deworming should be accompanied by water, sanitation, and hygiene (WASH) programming.

During interviews, stakeholders discussed whether policy changes would be needed in each state to launch cMDA. Most stakeholders in Goa felt that a new STH policy for cMDA was not necessary since cMDA was considered a "natural extension" or scale up of existing NDD programs. In Sikkim and Odisha, stakeholders suggested that a revised national policy would be necessary for launching cMDA and felt it would strengthen the legitimacy of the intervention. All stakeholders described similar processes for soliciting approval from key decision makers and decision-making bodies including, but not limited to, the Secretary of Health & Family Welfare, Principal Secretary, cabinet members, and ministers and the Chief Minister. A stakeholder from Goa explained, however, that policy processes can differ depending on whether program funding comes from national or state levels. Stakeholders communicated that strong, positive research findings can quickly garner political buy in and influence rapid approvals.

> "If we implement and show that there is reduction in worm infections and anemia, . . . [we] might take up [cMDA] as a state initiative and incorporate it as a routine activity." (Participant #6, Sikkim)

### Leadership structure

Readiness within the domain of leadership structure was high in all states, indicating effective leadership is in place for transitioning to cMDA for STH (median readiness score: 4–5, IQR:1) (Table 3). National MOHFW leadership was perceived to be generally receptive to new ideas or pilot projects (median readiness score: 5, IQR: 1). Stakeholders were comfortable in presenting new ideas or providing feedback on program implementation to their supervisors and communicated that it is a common practice. In Goa and Sikkim, stakeholders indicated that NTD program leadership at national, state, and district levels could effectively implement cMDA programs (including but not exclusively LF), (median readiness scores: 4.5–5, (IQR: 1). This score was lower for district-level leadership in Odisha (median readiness score: 2.5, IQR: 1.5).

In program maps, the highest concentration of activities for both LF and NDD programs were at the state level (Table 2). A major planning activity for both programs was the joint meeting convened at the state level, which included district-level leadership from across the state. Although housed in different departments, NVBDCP leadership also typically joined

**Table 3. Median and interquartile range for responses to organizational readiness survey.**

| Question | Goa (n = 6) Median (IQR) | Odisha (n = 6) Median (IQR) | Sikkim (n = 5) Median (IQR) |
|---|---|---|---|
| **Policy Environment** | | | |
| In my experience, India's national neglected tropical disease (NTD) policy supports implementation of community-wide mass drug administration (MDA). | 5 (1.5) | 5 (1) | 4.5 (0.5) |
| In my experience, the National NTD Implementation Plan is currently being implemented in my state as intended. | 5 (0.75) | 5 (0) | 3.5 (1.5) |
| I have observed that India's National NTD Implementation Plan provides sufficient guidance for implementing community-wide MDA programs, such as lymphatic filariasis (LF). | 5 (0) | 4 (1) | 4.5 (1) |
| I have observed that there is a collaborative network of external stakeholders (e.g., NGOs or technical/ financial partners) that would support community-wide deworming for STH in my state. | 4 (2) | 5 (1) | 5 (0) |
| I believe that my state needs to eliminate the transmission of STH. | 5 (0) | 5 (0.5) | 5 (0) |
| I have observed that my co-workers generally believe that my state needs to eliminate the transmission of STH. | 5 (0.75) | 3 (2) | 5 (0.25) |
| I believe that community-wide deworming can eliminate the transmission of STH in my state. | 4 (0.75) | 4 (0.5) | 5 (1) |
| I have observed that my co-workers generally believe that community-wide deworming can eliminate STH transmission in my state. | 4 (1) | 3 (2) | 5 (0.5) |
| I am supportive of implementing community-wide deworming for STH in my state. | 5 (0) | 5 (0.25) | 5 (1) |
| In my opinion, my co-workers will be supportive of implementing community-wide deworming for STH. | 5 (1) | 5 (2) | 5 (1) |
| Community-wide deworming for STH is not necessary for my state | 5 (0.75) | 5 (0) | 5 (0) |
| Community-wide deworming will not be able to stop transmission of worms in my state. | 4.5 (1.75) | 4 (1) | 4 (3) |
| I believe that MOHFW personnel within my state can deliver community-wide deworming with high coverage. | 5 (0) | 5 (0) | 5 (0) |
| I believe that my state can adapt the existing school deworming program (NDD) for community-wide deworming without too much difficulty. | 4.5 (1.5) | 5 (0) | 5 (0.25) |
| *Policy environment score* | **5 (1)** | **5 (1)** | **5 (1)** |
| **Leadership Structure** | | | |
| In my experience, the NTD program leadership at the National level is effectively implementing community-wide MDA programs in India. | 4.5 (1) | 5 (0) | 4.5 (1) |
| In my experience, the NTD program leadership at the state-level is effectively implementing community-wide MDA programs. | 5 (0.75) | 5 (0) | 4.5 (1) |
| In my experience, the NTD program leadership at the district level is effectively implementing community-wide deworming programs in India. | 5 (1) | 2.5 (1.25) | 4.5 (1.5) |
| In my experience, MOHFW leadership at the National level are generally receptive to new ideas or pilot projects. | 5 (1) | 5 (0) | 5 (0) |
| How often do your supervisors generally feel comfortable receiving feedback and recommendations from you or your colleagues on how to improve program implementation? | 4 (0.75) | 4 (2) | 4 (1) |
| How often do you present new ideas to your supervisor? | 5 (0.75) | 4 (1) | 4 (1) |
| It is challenging to present new ideas to my supervisor. | 3 (0.75) | 3 (0) | 3 (0.25) |
| How often do your subordinates generally feel comfortable providing feedback and recommendations to you or your colleagues on how to improve program implementation? | 4 (0.75) | 4 (0) | 4 (0) |
| Ministry of Education personnel that I work with on school or child interventions will likely support community-wide deworming for STH. | 5 (0) | 5 (2) | 5 (0) |
| The current implementation context within my state would negatively affect the ability of the state NTD program to deliver community-wide deworming for STH | 5 (1) | 5 (0) | 4 (0.25) |
| *Leadership structure score* | **4 (1)** | **5 (2)** | **4 (1)** |
| **Human Resources** | | | |
| My state will need additional training of NTD personnel to effectively deliver community-wide deworming for STH. | 1 (0) | 1 (0.25) | 1 (0) |
| Additional supervisors are needed within the state to coordinate the delivery of community-wide deworming for STH. | 1 (0) | 1 (3) | 1 (0) |

*(Continued)*

**Table 3.** (Continued)

| Question | Goa (n = 6) Median (IQR) | Odisha (n = 6) Median (IQR) | Sikkim (n = 5) Median (IQR) |
|---|---|---|---|
| There is low motivation amongst state NTD personnel to implement community-wide deworming for STH. | 4 (3) | 2 (1) | 2.5 (1.5) |
| I believe that NTD personnel in my state have the skills needed to implement a new community-wide deworming program. | 4.5 (1) | 3.5 (1.25) | 5 (1) |
| *Human resources score* | **2 (3)** | **2 (3)** | **1 (2.5)** |
| **Technical Capacity** | | | |
| In my experience, there is an effective program in my state for training drug distributors on how to deliver community-wide deworming. | 5 (0.75) | 5 (0.5) | 5 (1) |
| In my experience, NTD personnel within the state have demonstrated that they can deliver other community-wide MDA programs (e.g., lymphatic filariasis, LF) with high coverage. | 5 (0) | 2 (1.5) | 5 (0.25) |
| How often are treatment data incorrectly recorded during implementation of community-wide MDA programs? | 3 (1) | 3 (1) | 2.5 (1.25) |
| *Technical capacity score* | **5 (2)** | **3.5 (2.75)** | **4 (2)** |
| **Material Resources** | | | |
| My state has the key resources necessary to effectively implement community-wide deworming for STH. | 5 (0) | 4 (0.5) | 5 (0) |
| I have observed that deworming medicines are distributed from the state-level to local levels without too much difficulty. | 5 (0) | 5 (1) | 5 (0) |
| How often have you observed delays in the arrival of drugs for MDA programs due to supply chain problems? | 4 (2) | 3 (1) | 3 (0) |
| My state currently has the resources and tools needed to develop high-quality sensitization and education materials for community-wide deworming for STH. | 4.5 (1.75) | 2 (2.25) | 4 (2) |
| *Material resources score* | **5 (1)** | **4 (1)** | **4 (2)** |
| **Financial Resources** | | | |
| How often have you encountered difficulty in moving funds within the state level for a community-based program? | 3 (2.25) | 3.5 (1.25) | 2 (1) |
| How often have you observed difficulties with having enough funding from the National level to support implementation of community-based programs in your state? | 3.5 (1.75) | 3.5 (1) | 3 (1) |
| How often do you encounter difficulties with having enough funding at the district level to implement community-based programs? | 4 (1) | 2 (0) | 2 (1) |
| In my experience, drug distributors are given sufficient financial and/or non-financial incentives for administering community-wide MDA. | 4 (2.5) | 1 (0) | 3.5 (1.5) |
| I am not worried about whether my state has sufficient future funding for community-wide MDA programs. | 3.5 (3.25) | 1.5 (0.5) | 2 (0) |
| *Financial resources score* | **4 (2)** | **2 (1.75)** | **2 (1)** |
| **Community Delivery Infrastructure** | | | |
| I know of at least one community health program that could be used to deliver community-wide deworming for STH. | 5 (0) | 3 (3) | 5 (0.25) |
| In my experience, local drug distributors have the skills to effectively implement community-wide deworming for STH. | 2 (3) | 4 (1) | 5 (1) |
| How often are community members in your state resistant to community-wide MDA programs? | 3 (1) | 3 (1) | 3.5 (1.25) |
| *Community delivery infrastructure score* | **4 (2.25)** | **3 (2)** | **4 (1)** |

NDD coordination meetings to provide updates on LF MDA implementation. There were differences in the timing of this coordination meeting, with the LF meeting occurring five to six months prior to MDA as opposed to the NDD coordination meeting occurring two months prior to NDD (Table 2). These findings indicated that systems are already in place to coordinate between NDD and cMDA leaders, but modest changes to existing LF infrastructure may be needed to accommodate current NDD implementation norms.

In key informant interviews, stakeholders described the importance of also engaging local leaders and community groups. A list of important partners and leaders to engage are included in Annex 4 in S1 File.

*"I need to involve the community leaders first. . .I need them to give the talks before the health person comes. . .I need to convince these people first. So, sort of an advocacy with the people first, ok."* (Participant #2, Goa)

*"So, when you enter a new village, you have to talk to the leaders. . .. When they are convinced, they will take it up. . . then, you can make a village-level meeting."* (Participant #1, Odisha)

## Human resources

Readiness scores within the human resource domain were the lowest scores relative to other domains (median readiness score: 1–2, IQR: 0.5) (Table 3). Stakeholders from Goa and Sikkim had stronger confidence that their state has the skills needed to implement cMDA for STH (median score: 4.5, IQR: 1 and median score: 5, IQR: 1 respectively) compared to stakeholders from Odisha (median score: 3.5, IQR: 1.25). Stakeholders from Goa perceived a higher level of motivation amongst state NTD personnel (median score: 4, IQR: 3) than Odisha and Sikkim (median readiness scores 2 and 2.5 respectively). However, stakeholders from all states indicated that they did not have enough supervisors at the state level (median readiness score: 0–1, IQR: 0.25), and needed additional training of NTD personnel to effectively deliver cMDA for STH (median readiness score: 1, IQR: 3).

In program mapping, drug delivery in the LF program was led by ASHAs and volunteer community drug distributors (CDDs) in Odisha, and by multipurpose health workers in Goa. There was more overlap between the responsibilities of LF and NDD drug distribution personnel in Odisha, as compared to Goa (Table 2). Both LF and NDD programs utilize state-level supervision during drug distribution. State Monitoring & Evaluation (M&E) personnel made field monitoring visits to observe treatment delivery and helped to take corrective action if needed. For NDD, the monitoring visits occurred in schools and health centers, whereas in LF they occurred during home visits.

During key informant interviews, stakeholders believed that STH cMDA was labor intensive as it required monitoring of treatment compliance in addition to treating a larger target population. They shared concerns about human resource shortages when delivering cMDA activities. As more programs depended on AWWs and ASHAs, their workload had increased leading to low motivation levels, particularly for ASHAs who received incentives but were not salaried.

*"Government of India has so many programs, my staff is really overloaded. . .. In June we have the 'Intensive Diarrhea Control Fortnight' then comes the NDD then Pulse Polio. . . other programs like the noncommunicable diseases. . .. So, round the year there is something or the other. So, there is nothing like a separate staff, it is multitasking. . . that is why the coverage is not achieved because. . . we have to do many things at a time."* (Participant #1, Goa)

During key informant interviews, stakeholders from Goa discussed the need for more male staff. In a primarily urban state, respondents felt male workers would be needed for security reasons, to work late hours in order to reach residents when they are likely to be home.

*"If you go to the urban areas, you will find most of the houses closed. You can get them only after 6 (pm) so if you are employing staff, they will have to work at odd hours and they have to*

*be preferably male because, in Goa, no lady works after 6. . .. It's not safe."* (Participant #2, Goa)

## Technical capacity

Readiness driven by technical capacity to effectively implement cMDA for STH varied across the three states (median readiness score: Goa–5 (IQR:2), Sikkim–4 (IQR: 2), and Odisha–3.5 (IQR:2.75) (Table 3). Most stakeholders indicated their state has an effective drug distributor training program. Stakeholders from Goa and Sikkim indicated higher confidence that their state had demonstrated the ability to deliver cMDA with high coverage (median readiness score: 5, IQR: 0 and 0.25, respectively) than stakeholders from Odisha (median readiness score: 2, IQR: 1.5). In both readiness surveys and qualitative interviews, stakeholders from all states were concerned about accuracy of treatment data recorded during cMDA programs (median readiness scores: 2.5–3, IQR: 0.25). This was in part because teachers in the NDD program observed students taking the tablet, which made data recording a simpler activity for monitoring coverage. Stakeholders stressed that CDDs need additional capacity building, such as data recording and sensitization training, to effectively deliver STH cMDA.

*"We need lots of manpower to go to community. . . all health workers should be well trained to answer whatever questions people are asking,. . . they need to be sensitized thoroughly. . . also in interpersonal communication."* (Participant #3, Sikkim)

## Material resources

All three states exhibited high readiness in access to key material resources needed to implement STH cMDA (median readiness score: 5, IQR: 1) (Table 3). Stakeholders indicated that more resources will be needed for STH cMDA as compared to school based distribution, particularly for effective drug supply chains, training materials, informational databases/reporting systems, and community sensitization materials. Odisha indicated low readiness to develop high quality community sensitization materials for STH cMDA (median readiness scores: 2, IQR: 2.5). One stakeholder from Odisha attributed this to Odisha's vast diversity, with 22% of the population being tribal with over 25 dialects.

Stakeholders also highlighted supply chain challenges for distribution of material resources. Stakeholders from Odisha and Sikkim reported observing delays in drug arrival (median score: 3, IQR: 1) and challenges in supply chains (median readiness scores: 3, IQR: 1 and 0 respectively) that compromised their readiness to launch cMDA. Particularly in Sikkim, delays were exacerbated by the need to travel difficult terrain.

*"There are challenges because we have to depend upon Government of India for drug supply. . . the challenge is the procurement. . . so, the procurement process should start at least six months before . . .and at least two months before, the drugs should be in the state or district."* (Participant #2, Odisha)

Additionally, stakeholders from Goa and Sikkim discussed challenges estimating drug requirements due to in migration; stockouts were likely to increase following a transition to cMDA. Program mapping of LF cMDA in Goa and Odisha revealed that drugs are stored at the state level and distributed to district and block levels approximately two weeks before the MDA (Table 2). However, stakeholders in all states were concerned about where additional drugs could be stored since cMDA requires a much larger drug supply.

*"Of course, getting the logistics will be a challenge. Getting those tablets on time and storing them; distribution on the time. . . Sufficient space will not be there. So much stock will arrive for the whole community."* (Participant #3, Goa)

### Financial resources

The degree to which stakeholders were ready to launch cMDA was also affected by perceived access to financial resources, particularly in Odisha and Sikkim (Fig 1). The mean readiness score for financial resources in these two states was 2 (IQR: 1.75 and 1 respectively) (Table 3). Goa demonstrated a higher mean readiness score of 4 for financial resources (IQR: 2) and all states indicated concerns about the availability of funds within the state for cMDA for STH (mean readiness score: 1.5–3, IQR: 3).

State program maps demonstrated that cMDA for STH would not necessarily require extensive hiring of CDDs, as many were already working in both LF and NDD programs. However, key informants stressed that CDDs may require greater incentivization with such a large increase in workload. This converged with findings from readiness surveys where, for example, stakeholders in Odisha believed that CDDs may not have sufficient financial and nonfinancial incentives to engage in cMDA (median score: 1, IQR: 0). Almost every key informant discussed the urgent need to increase CDD pay and/or incentives, including suggestions for increased funding for cMDA from the national level.

*"They (drug distributors) are overburdened. They will do it but. . . we will have to motivate them. That's why I say funding . . . some incentive must be there for them otherwise program will not function. I mean, we won't be able to achieve the results."* (Participant #4, Goa)

Other costs associated with launching cMDA noted in individual interviews included developing and distributing sensitization materials and using more vehicles to reach all communities for treatment and monitoring visits. Sikkim and Odisha reported particularly low readiness to conduct these activities due to not enough funding at the district level (median readiness score: 2, IQR: 1). Lastly, all states indicated similar moderate readiness in their ability to move funds within the state for community-based programming (median score: 2–3.5, IQR: 1.25).

*"Ah, naturally it will be, cost for this community mobilization and all, lot more. . . . When your target increases. . ., the cost must expand. We do lot of IEC [information, education, and communication] work, BCC [behavior change communication], social mobilization, so fund is required, you have to give them their honorarium and other things."* (Participant #1, Odisha)

### Community delivery infrastructure

The program maps demonstrated significant overlap between community-level NDD and LF activities from the initial steps of estimating drug requirements, receiving implementation letters from MOHFW, and drug procurement, to the final steps of adverse event management and conducting monitoring visits (Table 2). The primary difference in activities was the personnel involved; NDD activities were primarily implemented by schoolteachers and AWWs whereas LF activities were led by frontline health workers (ASHA, AWWs, and multipurpose health workers).

During individual interviews, stakeholders in Goa and Odisha discussed leveraging existing LF community infrastructure for delivery of cMDA. They described challenges with this

approach because LF endemicity was mapped at the district level and MDA was not implemented uniformly across a state, however, other community health programs, such as Pulse Polio, immunization programs, Intensified Diarrhea Control Fortnight (IDCF), Village Health Nutrition Days, MCH programs, and noncommunicable disease control programs were believed to be a better fit for STH cMDA integration. While most stakeholders discussed integrated community-wide programs, some stakeholders from Odisha and Goa indicated that STH cMDA should be standalone and not build off an existing program to give it greater attention during implementation.

> *"We will have to do it in a mission mode . . .. and then target in next 10 days or 15 days and finish it. It has to be that way. . . make it as a separate program, we cannot integrate it my opinion. . . we are targeting different age groups from different backgrounds, people are working at different shifts—night shifts, day shifts."* (Participant #8, Goa)

Some stakeholders indicated STH cMDA could be delivered via fixed point treatment (out of schools, health centers, or community meeting spaces), however, others noted challenges with this approach. The majority felt that household distribution was more likely to be effective, with 1–3 mop up days. Some suggested adding door to door distributions while also continuing fixed point school and health center distribution.

> *"Best method is house to house. Booth [fixed point] approach was appropriate for Polio. It was a very long program and people already know the benefit. We tried booth approach one to two times for LF, we totally failed, no one came."* (Participant #1, Odisha)

In readiness surveys, Goa and Sikkim exhibited higher readiness in the community delivery infrastructure domain (mean readiness score: 4, IQR: 2.25 and 1 respectively) than Odisha (mean readiness score: 3, IQR: 2) (Table 3). All three states indicated that readiness to launch cMDA could be compromised by community member resistance to mass deworming (median score: 3–3.5, IQR:0.25). In individual interviews, stakeholders explained that communities can be skeptical of government interventions, believing they are substandard. They recommended that drug distributors wear gloves for improved sanitation and use consistent drug packaging. They described community members' concerns about side effects and the need for sufficient adverse event response management as an important component of community delivery infrastructure. Some examples of misinformation leading to treatment resistance included: perceptions that tablet should be taken on a full stomach, STH infections originate from eating sweets and meat, deworming treatment is needed only when worms are visible in the stool, and albendazole is contraindicated when taking other drugs. While misinformation was a key issue in all states, the types of misconception varied by state. For example, stakeholders from Goa discussed their state's high literacy leading to individuals seeking deworming information on the internet with varying accuracy, and independently seeking deworming medications from nongovernmental sources.

> *"There are also rumors. . . person suffering from diabetes, hypertension, they will not take drugs. So, there is a major miscommunication. Now fifty percent people are suffering from diabetes, blood pressure. . . They can also take albendazole and DEC, but they will not take."* (Participant #2, Odisha)

Stakeholders expressed the need for high quality community sensitization to combat misinformation. A list of community sensitization materials highlighted by stakeholders is included in Annex 5 in S1 File.

## Discussion

The current WHO and national guidelines for STH focus on the elimination of STH as a public health problem [11]. The transition to a STH transmission interruption strategy through cMDA would require systems redesign, including updating country-level NTD Master Plans, government budgets, partnership agreements, drug supply chains, and human resource plans [15]. While results from a cluster randomized trial testing the feasibility of STH transmission interruption (Deworm3 study) are pending from the state of Tamil Nadu (and other sites in Benin and Malawi), we studied opportunities and challenges to updating STH policy in India to help inform the potential rollout of an expanded treatment program [14,24].

Overall, all three states included in this multi-method study indicated a highly favorable policy environment, effective leadership structure, adequate material resources, demonstrated technical capacity, and adequate community infrastructure needed to launch a STH cMDA program. Availability of trained human resources in all the three states and financial resources in Odisha and Sikkim were indicated as currently inadequate for effective delivery. Similar findings were observed in readiness surveys in Tamil Nadu, India [21]. Stakeholders noted challenges such as workforce shortages, high workloads, and low motivation for treatment delivery. Incentives, community recognition, and supervisors' support have also demonstrated success as effective motivation for community drug distributors [25].

Stakeholders were skeptical that cMDA could eliminate STH due to lack of WASH access and risk of reinfection, yet there was notable commitment to implement cMDA for the purposes of anemia reduction, an STH-associated morbidity. Should a policy change occur, further aligning STH goals and operational guidelines with comorbidities (such as anemia and sanitation related diseases) could ensure that new programs are a high political priority.

Changes in organization policy and process are likely to be needed at all levels of government to shift from NDD to cMDA. Stakeholders explained that process changes can be expedited when strong evidence of effectiveness is presented to decision makers. These findings highlight the importance of dissemination of trial findings to STH policymakers at a national level. Additionally, many stakeholders stressed the importance of gaining consensus from local and religious leaders prior to launching cMDA and their inclusion in 'coordination committees'. Mechanisms need to be built across all levels of leadership to ensure that local leaders are not only consulted prior to implementation but are further engaged in planning [26,27].

Overall, Goa had highest readiness for implementing STH cMDA, followed by Odisha and Sikkim. The states reported both common and unique challenges to launching STH cMDA. Human resources were the most common challenge across states. Evidence from LF MDA programs in Mali, Nigeria, and Sierra Leone indicates that low motivation and an insufficient number of drug distributors can negatively influence treatment coverage [28]. Differential incentives across disease programs may also lead to low morale or disinterest in supporting a specific campaign [28]. However, in some settings, CDD motivation may have less to do with financial incentives, and instead be more greatly influenced by cultural, health system, and community related barriers [25].

Unique state-level challenges to launching cMDA for STH include that Goa would need to hire and train male CDDs due to safety concerns in urban areas. In Sikkim, stakeholders expressed concerns about the drug supply chain and vehicle expenses in Sikkim's hilly terrain. In Odisha, stakeholders indicated potential challenges in the drug supply chain, district-level leadership, incentivizing and motivating NTD personnel for cMDA, and bolstering resources to develop quality sensitization materials. Concerns about side-effects and misinformation about STH and its treatment was an issue in all the states. The participants, however, did not discuss potential emergence of drug resistance as being a challenge to implementing cMDA

for interrupting STH transmission. While well described in veterinary medicine, the clinical relevance of drug resistance markers to benzimidazoles detected in human STH infections is unclear but remains a potential risk of cMDA [29,30]. The states would be required to address all these specific challenges to achieve high cMDA coverage for STH. There is extensive evidence that sensitization materials should be contextually relevant, address local knowledge and beliefs, and that high quality sensitization materials can increase coverage up to fivefold [25, 28, 31].

This study investigated opportunities to integrate STH with LF programs and to build upon LF programs that have successfully achieved elimination and are in the process of de-implementation. Through triangulation of program mapping, readiness surveys, and key informant interviews it was evident that there are significant overlaps between NDD and LF campaigns, and LF platforms provide numerous opportunities for launching STH cMDA. Specific LF efforts that can be leveraged include drug procurement processes, supply chain management, and community delivery infrastructure. A systematic review similarly found that integrating NTD (onchocerciasis, schistosomiasis, and trachoma) program activities with LF MDA streamlined coordination, resource distribution, and incentivization of CDDs across programs [28]. Leveraging existing LF infrastructure, including the workforce, provides an opportunity to maximize resources and co-benefits of high coverage. Rather than simply adopt a "co-delivery" strategy, purposeful integration or strategic de-implementation of LF programs provides an opportunity to avoid activity duplication and leverage the relevant strengths of each program [31–33]. Some stakeholders in this study advocated for STH cMDA to be a standalone program to have greater control of the delivery schedule and perhaps greater community awareness when implemented in a campaign mode. However, the cycle of LF transmission is considered inefficient as few infectious microfilariae present in the mosquito penetrate a human host whereas risk of STH transmission is high because spread of contamination of soil with parasitic eggs [34,35].

The results of this study may also be generalizable to other Indian states considering transitioning from NDD to cMDA. The process itself, of attempting to preemptively understand readiness and implementation processes, may also be replicable across Indian settings that are considering a transition from NDD to cMDA and seeking to identify context-specific opportunities and challenges to doing so. The states with lowest STH prevalence as well as those phasing out the LF program could consider the transition to cMDA. However, the results of this study could differ across states implementing school-based deworming, and there could be variations in implementing cMDA at district-level within a state depending on the district-level STH prevalence and readiness to implement cMDA.

## Strengths and limitations

This analysis has several strengths and limitations. The selection of states across geographic regions of India provides for broader understanding of the need, opportunities, and challenges in transitioning from school-based deworming to cMDA for STH. The three different data collection activities used in this study provided the opportunity to triangulate data and identify areas of strong evidence. While the study includes a relatively small sample size of state-level NTD personnel, saturation was reached across relevant state-level experts.

## Conclusion

Stakeholders from three Indian states indicated that there is moderate to high readiness to transition from a policy of STH control strategy by NDD to STH transmission interruption strategy by cMDA, provided there are new investments in financial, material, and human

resources for state-level programs. The transition of LF programs into a post MDA surveillance phase offers an opportunity to leverage existing LF infrastructure in a strategic de-implementation approach. Currently, leadership structures are separate for these two programs, however, the three participating states reported extensive experience with integration. Program maps further provide opportunities to visualize specific resources and activities that can be leveraged across programs. Thoughtful integration may offer the opportunity to maximize resources and launch successful STH transmission interruption programs at scale.

## Supporting information

**S1 File. Annexures.**
(DOCX)

## Acknowledgments

We thank the MOHFW and NITI Aayog, for their guidance and support in carrying out this study. We specially thank officials at the MOHFW of the state of Goa, Sikkim, and Odisha for giving their valuable time to undertake this landscaping exercise.

## Author Contributions

**Conceptualization:** Jayaprakash Muliyil, Judd L. Walson, Sitara S. R. Ajjampur, Arianna Rubin Means.

**Data curation:** Kumudha Aruldas, Kim Dawson, Malvika Saxena, Angelin Titus, Jabaselvi Johnson.

**Formal analysis:** Kim Dawson, Malvika Saxena, Marie-Claire Gwayi-Chore, Arianna Rubin Means.

**Funding acquisition:** Judd L. Walson.

**Investigation:** Angelin Titus.

**Methodology:** Ajay Khera, Sitara S. R. Ajjampur, Arianna Rubin Means.

**Project administration:** Sitara S. R. Ajjampur.

**Supervision:** Kumudha Aruldas.

**Visualization:** Kim Dawson, Arianna Rubin Means.

**Writing – original draft:** Kumudha Aruldas, Kim Dawson.

**Writing – review & editing:** Kumudha Aruldas, Malvika Saxena, Angelin Titus, Jabaselvi Johnson, Marie-Claire Gwayi-Chore, Jayaprakash Muliyil, Gagandeep Kang, Judd L. Walson, Ajay Khera, Sitara S. R. Ajjampur, Arianna Rubin Means.

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
