## [Decision Letter · Decision Letter 0]

31 Oct 2022

Dear Dr. Ajjampur,

Thank you very much for submitting your manuscript "Evaluation of opportunities to implement community-wide mass drug administration for elimination of soil transmitted helminths in India" for consideration at PLOS Neglected Tropical Diseases. As with all papers reviewed by the journal, your manuscript was reviewed by members of the editorial board and by several independent reviewers. In light of the reviews (below this email), we would like to invite the resubmission of a significantly-revised version that takes into account the reviewers' comments. 

We cannot make any decision about publication until we have seen the revised manuscript and your response to the reviewers' comments. Your revised manuscript is also likely to be sent to reviewers for further evaluation.

Sincerely,

Maria Angeles Gómez-Morales, PhD

Academic Editor

Eva Clark

Section Editor

Reviewer's Responses to Questions

**Key Review Criteria Required for Acceptance?**

**Methods**

-Are the objectives of the study clearly articulated with a clear testable hypothesis stated?

-Is the study design appropriate to address the stated objectives?

-Is the population clearly described and appropriate for the hypothesis being tested?

-Is the sample size sufficient to ensure adequate power to address the hypothesis being tested?

-Were correct statistical analysis used to support conclusions?

-Are there concerns about ethical or regulatory requirements being met?

Reviewer #1: Clearly explained

Reviewer #2: Objective and pragmatic result of the study are clearly stated. 

The population of study is clearly described throughout the methodology

Reviewer #3: - The authors should define "elimination". WHO aim is to eliminate Soil-transmitted helminths as a health problem, what is not synonim as eliminating all infections. 

- I suggest the inclusion of a figure contabilizing and displaying the number of participants as well as the category they belong. 

- The n of participants in Organizational Readiness Survey and in Qualitative Interviews should be displayed.

**Results**

-Does the analysis presented match the analysis plan?

-Are the results clearly and completely presented?

-Are the figures (Tables, Images) of sufficient quality for clarity?

Reviewer #1: well presented

Reviewer #2: Results are clearly and completely presented.

Reviewer #3: -Table 1 should specify what the acronyms mean, maybe as a footnote. A flow diagram chould be a complementary figure to this one.

**Conclusions**

-Are the conclusions supported by the data presented?

-Are the limitations of analysis clearly described?

-Do the authors discuss how these data can be helpful to advance our understanding of the topic under study?

-Is public health relevance addressed?

Reviewer #1: see general comments

Reviewer #2: The conclusions are supported by the data presented.

Reviewer #3: - The authors should further discuss the following: How represenative are these results? What is the n of the participants and their role? Do they have conflict of interest for cMDA? Did any biased could have occurred? 

- The authors should discuss the disadvantage of implementing cMDA, such as treatment resistance, already observed in few regions around the globe.

**Editorial and Data Presentation Modifications?**

Reviewer #1: (No Response)

Reviewer #2: (No Response)

Reviewer #3: -Table 1 should specify what the acronyms mean, maybe as a footnote. A flow diagram chould be a complementary figure to this one.

**Summary and General Comments**

Reviewer #1: I think the paper is interesting but need a comprehensive rewriting especially on the introduction and in the discussion as many of the statements made by the authors are not supported by evidence.

For example:

Line 33 he authors mention “evidence that may be possible to interrupt STH transmission” without bringing any support to this statement in any part of the paper; in reality there is no evidence that cMDA would interrupt transmission in areas where morbidity is relevant (and there are serious doubts if interrupting STH transmission in areas where there is no STH morbidity has any public health value).

Line 78 the author hypnotize that the same intervention that has been successful for lymphatic filariasis would be effective STH also this statement is not supported by any evidence, firstly it is not possible compare the transmission of LF (that is inefficiently transmitted by mosquito) with a very efficient STH transmission through environmental contamination; secondly not a single country (or area of a country) where STH morbidity was significant was ever able to interrupt transmission without an increase of the sanitation level that impede environmental contamination with human excreta. 

Line 442 cites WHO road map as source for “ increasing interested of using expanded MDA to interrupt STH “ this statement is not present in the WHO road map and therefore the statement is misleading

In conclusion I would suggest the authors to keep the focus on reporting the results of their investigation on the theoretical interest of local stakeholder in expanding the present target of deworming (the results are clearly presented) but avoiding to extrapolate on fantasies regarding the elimination of transmission that at the moment is not proved to be possible by drug alone in area of high endemicity.

In addition in my opinion in the "limitations of the study" two important issues should be mentioned:

1- the discussion about the financial coverage of the additional cost costs linked to cMDA has been conducted as additional resources could be made available from an higher level in MoH or donors. It would have been interesting to know if the stakeholder investigated would be ready to use part of their present budget to cover these additional cost (a positive attitude on this aspect would have been a more clear sign of feasibility).

2- The respondents were not informed about the additional risk linked to the expansion of the target group of stimulating drug resistance by increasing the coverage in individual that are at less risk of morbidity. The rick that their answers were not sufficiently informed should be mentioned.

Reviewer #2: This study objective is to determine the readiness of government organizations in three Indian states for transitioning from school-based MDA to cMDA. The prinicipal findings of the study indicate that the necessary conditions to implement this intervention are favorable in these three states. The methodology was thorough and the results are clearly explained. This type of studies offer insights on government's readiness to apply different interventions in order to aim for STH elimination and not only STH control.

Reviewer #3: This study is very relevant for the field. The manuscript is very well written and easy to follow. However, the representativeness of this results in this region should be further explained and discussed.

PLOS authors have the option to publish the peer review history of their article (what does this mean?). If published, this will include your full peer review and any attached files.

Reviewer #1: No

Reviewer #2: No

Reviewer #3: No
---

## [Decision Letter · Decision Letter 1]

24 Jan 2023

Dear Dr. Ajjampur,

Thank you very much for submitting your manuscript "Evaluation of opportunities to implement community-wide mass drug administration for interrupting transmission of soil-transmitted helminths infections in India" for consideration at PLOS Neglected Tropical Diseases. As with all papers reviewed by the journal, your manuscript was reviewed by members of the editorial board and by several independent reviewers. The reviewers appreciated the attention to an important topic. Based on the reviews, we are likely to accept this manuscript for publication, providing that you modify the manuscript according to the review recommendations. Please, carefully read the comments from reviewer 1

Sincerely,

Maria Angeles Gómez-Morales, PhD

Academic Editor

Eva Clark

Section Editor

Reviewer's Responses to Questions

**Key Review Criteria Required for Acceptance?**

**Methods**

-Are the objectives of the study clearly articulated with a clear testable hypothesis stated?

-Is the study design appropriate to address the stated objectives?

-Is the population clearly described and appropriate for the hypothesis being tested?

-Is the sample size sufficient to ensure adequate power to address the hypothesis being tested?

-Were correct statistical analysis used to support conclusions?

-Are there concerns about ethical or regulatory requirements being met?

Reviewer #1: see general comments

Reviewer #2: The study's objective is clearly indicated in the abstract but not so clearly in the text. I suggest adding a sentence at the end of the introduction that explains the main objective.

I believe the study design is appropriate for determining these regions' readiness to implement cMDA. 

It is difficult for a foreign reader to determine whether the regions chosen for study development have the highest STH prevalence.

How would the results of this study differ if it was conducted in more remote areas?

More information about the region's status should be included.

Reviewer #3: The comments and suggestions have been addressed.

**Results**

-Does the analysis presented match the analysis plan?

-Are the results clearly and completely presented?

-Are the figures (Tables, Images) of sufficient quality for clarity?

Reviewer #1: see general comments

Reviewer #2: The study sites' institutional readiness is clearly described throughout the results.

I find the results to be of adequate quality and clarity.

Reviewer #3: The comments have been addressed.

**Conclusions**

-Are the conclusions supported by the data presented?

-Are the limitations of analysis clearly described?

-Do the authors discuss how these data can be helpful to advance our understanding of the topic under study?

-Is public health relevance addressed?

Reviewer #1: see general comments

Reviewer #2: The data presented supports the authors' conclusions, and the findings are of great interest in order to provide knowledge of how effective a transition from school-based deworming to cMDA could be in an endemic region with prior cMDA infrastructure.

Reviewer #3: Regarding anthelmintic resistance, I would still suggest the authors to mention it during the dicussion. This is probably the main and more relevant drawback of cMDA. Moreover, it has been studied in different regions of the world. Thus, should be at least mentioned.

**Editorial and Data Presentation Modifications?**

Reviewer #1: (No Response)

Reviewer #2: (No Response)

Reviewer #3: The comments have been addressed.

**Summary and General Comments**

Reviewer #1: The review made by the authors is in my opinion acceptable except for two points in the introduction that are essential to put the results obtained in a context:

Line 58-61 The authors do not mention that the main reason why cMDA is not recommended by WHO is to avoid the development of drug resistance against the only presently available drugs. Parasite residence against albendazole and mebendazole is well documented in veterinary medicine where the entire population of infected animal has been treated for several years (1)

Line 78 80 The authors, mention that cMDA was successful in interrupting LF transmission without mentioning that the LF transmission cycle is relatively inefficient (2) (microfilariae are transported by mosquito) while STH transmission is extremely efficient (3) (contamination of environment with parasite eggs)

References 

1 Coles, G.C., et al 2006. The detection of anthelmintic resistance in nematodes of veterinary importance. Vet. Parasitol. 136, 167–185.

2- Streit T, Lafontant JG. Eliminating lymphatic filariasis: a view from the field. Ann N Y Acad Sci. 2008; 1136:53-63. 

3- Horiuchi S, Paller VG, Uga S. Soil contamination by parasite eggs in rural village in the Philippines. Trop Biomed. 2013; 30(3):495–503.

Reviewer #2: (No Response)

Reviewer #3: The comments have been addressed.

PLOS authors have the option to publish the peer review history of their article (what does this mean?). If published, this will include your full peer review and any attached files.

Reviewer #1: No

Reviewer #2: No

Reviewer #3: No

Figure Files:

Data Requirements:

Reproducibility:

References

---

## [Editor Report · Decision Letter 2]

15 Feb 2023

Dear Dr. Ajjampur,

We are pleased to inform you that your manuscript 'Evaluation of opportunities to implement community-wide mass drug administration for interrupting transmission of soil-transmitted helminths infections in India' has been provisionally accepted for publication in PLOS Neglected Tropical Diseases.

Best regards,

Maria Angeles Gómez-Morales, PhD

Academic Editor

Eva Clark

Section Editor

---

## [Editor Report · Acceptance letter]

24 Feb 2023

Dear Prof. Ajjampur,

We are delighted to inform you that your manuscript, "Evaluation of opportunities to implement community-wide mass drug administration for interrupting transmission of soil-transmitted helminths infections in India," has been formally accepted for publication in PLOS Neglected Tropical Diseases.

Best regards,

Shaden Kamhawi

co-Editor-in-Chief

Paul Brindley

co-Editor-in-Chief
